# Modulation of Endocannabinoids by Caloric Restriction Is Conserved in Mice but Is Not Required for Protection from Acute Kidney Injury

**DOI:** 10.3390/ijms22115485

**Published:** 2021-05-22

**Authors:** Karla Johanna Ruth Hoyer-Allo, Martin Richard Späth, Ruth Hanssen, Marc Johnsen, Susanne Brodesser, Kathrin Kaufmann, Katharina Kiefer, Felix Carlo Koehler, Heike Göbel, Torsten Kubacki, Franziska Grundmann, Bernhard Schermer, Jens Brüning, Thomas Benzing, Volker Burst, Roman-Ulrich Müller

**Affiliations:** 1Department II of Internal Medicine and Center for Molecular Medicine Cologne, University of Cologne, Faculty of Medicine and University Hospital Cologne, Kerpener Str. 37, 50937 Cologne, Germany; johanna.hoyer-allo@uk-koeln.de (K.J.R.H.-A.); martin.spaeth@uk-koeln.de (M.R.S.); marc.johnsen@cellitinnen.de (M.J.); felix.koehler@uk-koeln.de (F.C.K.); torsten.kubacki@uk-koeln.de (T.K.); franziska.grundmann@uk-koeln.de (F.G.); bernhard.schermer@uk-koeln.de (B.S.); thomas.benzing@uk-koeln.de (T.B.); 2Cologne Excellence Cluster on Cellular Stress Responses in Aging-Associated Diseases (CECAD), University of Cologne, Faculty of Medicine and University Hospital Cologne, Joseph-Stelzmann-Straße 26, 50931 Cologne, Germany; susanne.brodesser@uk-koeln.de (S.B.); k.kaufmann1@gmx.net (K.K.); katharinak21@gmail.com (K.K.); jens.bruening@uk-koeln.de (J.B.); 3Max Planck Institute for Metabolism Research, Gleueler Str. 50, 50931 Cologne, Germany; ruth.hanssen@uk-koeln.de; 4Policlinic for Endocrinology, Diabetes and Preventive Medicine (PEPD), University of Cologne, Faculty of Medicine and University Hospital Cologne, Kerpener Str. 62, 50937 Cologne, Germany; 5Institute of Pathology, University Hospital of Cologne, Kerpener Str. 37, 50937 Cologne, Germany; heike.goebel@uk-koeln.de

**Keywords:** acute kidney injury, ischemia-reperfusion injury, preconditioning, caloric restriction, stress resistance, endocannabinoid, anandamide

## Abstract

Acute kidney injury (AKI) is a frequent and critical complication in the clinical setting. In rodents, AKI can be effectively prevented through caloric restriction (CR), which has also been shown to increase lifespan in many species. In *Caenorhabditis elegans* (*C. elegans*), longevity studies revealed that a marked CR-induced reduction of endocannabinoids may be a key mechanism. Thus, we hypothesized that regulation of endocannabinoids, particularly arachidonoyl ethanolamide (AEA), might also play a role in CR-mediated protection from renal ischemia-reperfusion injury (IRI) in mammals including humans. In male C57Bl6J mice, CR significantly reduced renal IRI and led to a significant decrease of AEA. Supplementation of AEA to near-normal serum concentrations by repetitive intraperitoneal administration in CR mice, however, did not abrogate the protective effect of CR. We also analyzed serum samples taken before and after CR from patients of three different pilot trials of dietary interventions. In contrast to mice and *C. elegans*, we detected an increase of AEA. We conclude that endocannabinoid levels in mice are modulated by CR, but CR-mediated renal protection does not depend on this effect. Moreover, our results indicate that modulation of endocannabinoids by CR in humans may differ fundamentally from the effects in animal models.

## 1. Introduction

Acute kidney injury (AKI) is defined as an acute decline in kidney function, leading to an increase in serum creatinine and/or a decrease in urinary excretion [1]. It is one of the most frequent complications in hospitalized patients. Particularly in the elderly, AKI shows a rapidly rising incidence and is strongly associated with increased mortality and morbidity [2]. Although kidney function tends to recover at least partially, AKI is associated with a deterioration in clinical outcomes including an increased risk of chronic kidney disease (CKD) and progression to end-stage renal disease (ESRD) [3,4,5,6,7,8]. In general, progression to ESRD can be the result of repetitive, minor events of AKI during a lifetime [9,10]. While pre- and post-renal causes of AKI can be corrected rapidly in most of the cases, there is no causal therapy for intrarenal AKI (e.g., due to ischemia-reperfusion injury (IRI) or drug toxicity) in the clinical setting [11,12,13,14,15,16,17].

Despite knowledge of the risk factors and decades of research on AKI, targeted approaches toward prevention are still missing. In contrast, extremely effective approaches have been developed that are capable of preventing or attenuating AKI in rodent models [18,19,20,21]. These preconditioning strategies enhance cellular stress resistance before exposure to the injurious stimulus. Caloric restriction (CR) is one of the most potent preconditioning protocols and has been shown to increase stress resistance and increase lifespan in a multitude of species ranging from the nematode *Caenorhabditis elegans* (*C. elegans)* to non-human primates [19,20,21,22,23]. However, the effect of CR in humans on stress resistance is less clear. Clinical trials in this context are complicated by a number of factors. First, patient cohorts at risk are quite heterogenous and show high degrees of co-morbidity. Knowledge on normal calorie intake in this population is very limited. Furthermore, blinding of CR is difficult and this fact can lead to undesired dietary adaptation in the control group [24,25,26]. Taken together, an improved knowledge of the molecular mechanisms underlying the impact of CR on resilience is an important pre-requisite for clinical translation of its potential. Interestingly, CR has been shown to result in a significant reduction of N-acylethanolamines (NAEs) in *C. elegans.* This decrease of NAEs was required and sufficient for the CR-mediated extension of lifespan [27]. Consequently, the hypothesis that the reduction of NAEs upon CR may also be involved in organ protection in mammals is an important research question. NAEs are endocannabinoids and as such lipid-derived signaling molecules that are, among other organs-synthesized in the kidney [28,29,30]. They are formed within the tissues by N-acyl-ethanolamine phospholipids (NAPE) catalyzed by a membrane-associated NAPE-hydrolyzing phospholipase D (NAPE-PLD) [31]. NAEs can activate the G-protein-coupled receptors CB-1 and CB-2 in the central nervous system and peripheral organs (e.g., the kidney) [32]. They are involved in the regulation of food intake [33], energy metabolism [34], inflammation [35], and vascular tonus [28].

The aim of this study was to evaluate whether (1) CR-induced reduction of NAE levels is conserved in mammals and (2) whether this reduction is involved in CR-induced renal organ protection. To reach this goal, we used a murine model of renal IRI combined with endocannabinoid supplementation to calorically restricted mice.

## 2. Results

### 2.1. Anandamide Is Reduced by CR and Can Be Increased by Intraperitoneal Administration in Mice

To examine if the CR-induced reduction of endocannabinoids previously observed in *C. elegans* was conserved in mice, we analyzed plasma samples of male C57Bl6J wildtype mice after CR and without intervention (Figure 1A). In *C. elegans*, among a row of different NAEs reduced by CR, the supplementation of two C20-fatty-acid-containing NAEs—eicosapentaenoyl ethanolamide (EPEA) and arachidonoyl ethanolamide (AEA)—was able to rescue phenotypes associated with NAE depletion [27]. Whilst EPEA is more abundant in worms, this NAE was not detectable in mouse serum (Figure 1A). However, AEA was the most abundant C20-fatty-acid containing endocannabinoid and strongly reduced by CR in rodents (Figure 1A,B).

Consequently, we decided to examine whether the reduction in AEA was required for CR-mediated renal organ protection using a mouse model of unilateral nephrectomy and contralateral IRI. As a first step, we performed a dose-finding experiment for AEA supplementation. Since the protective effects of CR can be seen already after a few days, we deemed a period of eight days of supplementation sufficient for this purpose (Figure 1C). Intraperitoneal injections of 0.00 (vehicle), 0.02, 0.05, 0.075, and 0.1 mg/g bodyweight resulted in a linear rise (r^2^= 0.96, *p* = 0.38) of AEA. Since already the lowest concentration resulted in a strong increase in AEA serum levels (Figure 1D), we continued the experiment using a concentration of 0.01 mg/g bodyweight.

### 2.2. The CR-Mediated Protection from Renal IRI Does Not Depend on the Reduction of AEA

To examine if CR-mediated protection from AKI depends on the loss of AEA, we injected AEA (0.01 mg/g bodyweight) or vehicle to calorically restricted mice (CR) or ad libitum fed controls (CTRL). All groups underwent renal IRI or sham-IRI (sham) simultaneously (Figure 2A).

Intraperitoneal supplementation of AEA or vehicle did not influence weight loss upon CR (Appendix A). As expected, 24 h after IRI, creatinine and blood urea nitrogen (BUN) were significantly increased in ad libitum fed mice (CTRL) compared to the sham operated animals (sham) (Figure 2B,C). This almost 15-fold increase was entirely prevented by CR (Figure 2B,C). The protective effect of CR was not affected by AEA supplementation (Figure 2B,C). Likewise, AEA or vehicle supplementation in ad libitum fed mice did not affect the outcome after IRI (CTRL_AEA/CTRL_veh, Appendix A).

### 2.3. Intraperitoneal AEA Supplementation Does Not Abrogate the CR-Mediated Protection from Cell Death and Tubular Damage

For histological examination of tubular injury periodic acid-Schiff-staining (PAS) was performed. As expected, kidneys of ad libitum fed mice (CTRL) showed increased pyknosis, stronger brush border loss and stronger epithelial flattening after IRI in comparison to sham-operated (sham) and to CR-mice (Figure 3).

Kidneys of CR-mice injected with AEA or vehicle (CR_AEA/CR_veh) were indistinguishable from CR animals without intraperitoneal injections (CR) after IRI on the level of histology. Likewise, supplementation of AEA or vehicle to ad libitum fed mice did not affect the histological outcome after IRI (CTRL_AEA/CTRL_veh). This was confirmed in a semi-quantitative analysis of the histological damage using a composite damage scoring system (Appendix A).

In addition, we assessed cell death in kidneys 24 h after IRI using cleaved-caspase 3 staining and terminal deoxynucleotidyl transferase dUTP nick end labeling (TUNEL). Whilst kidneys of the ad libitum-fed mice (CTRL) showed a strong increase in cell death, this increase was entirely prevented by CR. Again, AEA supplementation or vehicle administration did not have any impact on this protection (CR-AEA/CR-veh, Figure 3) and did not change the induction of cell death after IRI in AL-fed mice (CTRL-AEA/CTRL-veh) (Appendix A).

### 2.4. CR Is Associated with an Increase of AEA in Humans

To answer the question if CR-induced reduction of endocannabinoids and specifically AEA is even conserved in humans, we assembled a set of longitudinally acquired human serum samples before and after CR derived from different investigator initiated clinical trials at our center. The first trial investigated the protective effect of CR in patients undergoing elective cardiac surgery on cardiopulmonary bypass (CR_KCH, NCT01534364, [36]). After randomization, subjects received a formula diet that supplied 60% of daily energy expenditure (DEE) for the last seven days prior to surgery; control subjects were not restricted in their food intake. In contrast to the findings in mice and worms, analysis of serum samples of 19 patients (>60 years of age) revealed significantly higher AEA levels after CR (Figure 4A).

Consequently, we decided to check AEA in serum samples from a second study examining middle-aged (42–64 years, *n* = 6) healthy living kidney donors that were provided a formula diet containing 50% of the DEE for the last week prior to kidney donation (CR_LSP, NCT02745444) (Figure 4B). Again, CR led to a significant increase of AEA (Figure 4B).

Finally, samples from a third cohort (optifast) of obese individuals (BMI > 34 kg/m^2^, 29–65 years, *n* = 8) undergoing a 3-month formula diet with the fixed amount of 800 kcal/d to reach weight-loss showed a trend toward higher AEA levels after CR (*p*-value: 0.1296) (Figure 4C). Detailed baseline characteristics of the three cohorts are shown in Appendix A.

## 3. Discussion

CR-mediated preconditioning has been shown to be extremely effective in the prevention of AKI in rodents [17,19,22,37]. In *C. elegans* CR is a classical longevity intervention increasing lifespan and metabolic heath. In line with the effect on organ protection in mammals, longevity pathways also increase stress resistance in nematodes. Reduced N-acyl-ethanolamine (NAE) levels upon CR have been shown to be both sufficient and required for CR-mediated lifespan extension [27].

NAEs comprise a heterogeneous group of lipid-derived molecules that can activate the G-protein-coupled receptors CB-1 and CB-2 in the central nervous system and peripheral organs (e.g., the kidney) [32]. NAE signaling is involved in the regulation of food intake [33], energy metabolism [34], inflammation [35], and vascular tonus [28]. Nevertheless, the exact underlying mechanisms remain elusive [28].

CR led to a reduction of six different NAEs in the nematode, whilst only supplementation of the C20-fatty acid containing NAEs—EPEA and AEA—were able to prevent Dauer arrest and promote reproductive growth of *daf-2* mutants [27]. In the present study, we were able to show that CR significantly reduced NAEs including AEA in mice.

Notably, CR consisting of 80% of the daily intake for two weeks did not influence AEA levels of C57Bl6J mice in one previous study [38]. We assumed this difference to be the result of a more stringent CR protocol (i.e., a longer period of lower calorie intake in the study at hand).

EPEA is known to be more abundant in worms than in rodents [39], explaining why this NAE was not detected in our mouse serum. Instead, AEA was the C20-fatty-acid containing NAE most strongly reduced by CR in our experiment. Therefore, we decided to evaluate the impact of AEA-reduction on CR-induced renal organ protection in mice.

AEA is the first NAE identified [32,40]. Whilst there is ample evidence that AEA is generated by enzymatic cleavage of N-arachidonoyl-phosphatidylethanolamine (NAPE) through NAPE-phospholipase D [41,42,43,44], the underlying mechanism of NAPE-generation in the kidneys remains elusive [32]. AEA shows high abundance in renal tissue [40,45] and was detected in endothelial and mesangial cells [30]. AEA showed higher concentrations in the renal medulla than in the cortex [32,46]. Regarding its function, AEA can activate CB-1/2 receptors, but also transient receptor potential vanilloid type I channels (TRPV1) localized in the vasculature [32,47]. Whilst CB-1 is expressed nearly ubiquitously in the nephron [48], CB-2 was only detected in podocytes [49], proximal tubular [50], and mesangial cells [30]. AEA is inactivated enzymatically by fatty acid amide hydrolase (FAAH) and monoacylglycerol lipase (MGL) [51]. Under physiologic conditions, AEA contributes to the regulation of glomerular filtration, the vascular tonus, and sodium-transport [28].

Current mechanistic concepts suggest that CR-mediated cellular resistance is based on reduced IGF-1/Insulin signaling [19], activation of AMPK [52], and an inhibition of the mammalian target of rapamycin (mTOR) [53]. In the nematode, Lucanic et al. concluded that the TOR-pathway might control NAE levels in response to nutrient intake and that reduced NAE signaling is a component of CR-mediated lifespan extension [27]. On the other hand, there is evidence that NAEs activate mTOR [54,55] and AMPK signaling in renal proximal tubular cells [56] as well as hepatocytes [57].

In the context of AKI, the role of NAEs seems to be ambivalent. Whereas CB-1-inhibition reduced the decline of kidney function after cisplatin administration [58], CB-1 and CB-2 activation was demonstrated to mediate protective effects in renal IRI by increasing the renal blood flow through vasodilatation [59,60,61]. Furthermore, inhibition of the degradation of another NAE—of 2-Arachidonylglycerol (2-AG)—was shown to improve the outcome after bilateral renal IRI [62].

Here, we provide the first detailed study of CR-dependent NAE-modulation in mice and humans combined with pharmacological testing of the contribution of AEA to CR-mediated renal organ protection. In line with previous investigations, we found AKI induced by renal IRI to be completely prevented by CR in C57Bl6J mice. This protection was accompanied by the reduction in AEA. However repetitive intraperitoneal supplementation of AEA did not show any impact on the outcome 24 h after renal IRI (Figure 2 and Appendix A). It is important to emphasize a number of differences in study design compared to previously published work on NAEs in renal IRI. First, different models of renal IRI (uni-/bilateral without nephrectomy) were used [59,60,61,62]. Second, this is the first study employing repetitive intraperitoneal supplementation of AEA. Finally, differences between mouse strains may also play a role, since only Moradi et al. used C57Bl6 mice in a model of bilateral IRI [62].

Since the lack of effect of AEA-supplementation could have been a consequence of the experimental protocol used, we decided to—before moving on to additional phenotypic mouse studies—check whether AEA reduction upon CR was indeed also conserved in humans. This knowledge would also be important as a potential basis to human trials modulating NAE levels by CR.

Interestingly, the literature concerning dietary influences on AEA in humans is much more controversial than in rodents [38] or *C. elegans* [27]. Therefore, we analyzed human serum samples acquired before and after short-term CR in three independent cohorts. Of note, results from two of these cohorts revealed a significant increase in AEA after CR. The third study showed a trend toward this direction (*p* = 0.1296) not reaching statistical significance, possibly as a consequence of comparably high heterogeneity of this cohort. Notably, recent publications provide evidence for either no influence on AEA [63,64] or a reduction of AEA after longer-term interventions (e.g., gastric-bypass, life style interventions programs, or prolonged CR) [65,66,67]. Considering that fasting increases and semi-starvation reduces endocannabinoid levels [68,69], it seems possible that the observed differences are related to the duration of the particular interventions. The mechanisms underlying the impact of different CR protocols on NAE levels will be an interesting question for future studies.

Taken together, this study clearly shows that NAEs are reduced by CR in mice. Nevertheless, this effect does not seem to be involved in CR-mediated protection from murine renal IRI. Furthermore, our analyses indicate that the CR-induced modulation of NAEs in humans may differ fundamentally from the effects in animal models.

## 4. Methods

### 4.1. Animal Procedures

All animal experiments were carried out in accordance with the LANUV NRW (Landesamt für Natur, Umwelt und Verbraucherschutz Nordrhein-Westfalen, State Agency for Nature, Environment and Consumer Protection North Rhine-Westphalia; VSG 84–02.04.2013.A158). Male wildtype C57/Bl6J mice were provided by Charles River Laboratories (Wilmington, MA, USA) and were used for all experiments. All mice were kept under similar specific pathogen free (SPF) conditions in group-cages with up to five mice at the same time and a daily 12 h light and 12 h dark rhythm. The mice received ad libitum access to drinking water and food with one exception for the caloric restriction. The standard food chow was purchased from sniff GmbH (Soest, Germany).

As previously described [17,22,37], calorically restricted mice were provided with 66% (3 g per day per mouse) daily in the morning on the last 28 days prior to surgery. Before initiation of dietary preconditioning, CR-mice were set individually in new cages. Plasma samples of calorically restricted animals were obtained in the morning before feeding.

Anandamide was purchased by Cayman Chemical (Ann Arbor, MI, USA) and administrated (0.01 mg/g bodyweight) by daily intraperitoneal injections in a volume of 10 µl/g bodyweight for one week prior to IRI (11 to 12 weeks of age). Standard sodium-chloride solution (0.9%) with 0.05% ethanol in an amount of 10 µL/g bodyweight was used for the vehicle control group.

### 4.2. Ischemia-Reperfusion Injury (IRI)

Renal IRI surgery was performed in 12–13-week-old male C57Bl6J mice under anesthesia (ketamine (Zoetis, Berlin, Germany)/xylazine (Bayer, Leverkusen, Germany). To maintain the body temperature, surgery was performed on a temperature-controlled heating pad (Havard apparatus, Holliston, MA, USA). As described before [22,37], following midline abdominal incision, the vessels of the right kidney were ligated with sutures and the kidney was removed afterward. Following this, the left renal artery and vein underwent unilateral clamping for 40 min with an atraumatic micro-vascular clamp and subsequent removal of the clamp for reperfusion, which was controlled visually. The abdominal incision was closed in two layers by silk suture in the body wall and clips in the skin.

Sham-operated animals underwent the same approach with the exception that the left renal pedicles were not clamped. After surgery, the mice obtained normal saline containing 0.2 mg buprenorphine subcutaneously and access to drinking water supplemented with metamizole. To prevent mutilation of the wound by fellow mice, all operated mice were transferred to single cages.

Twenty-four hours after reperfusion, the mice received intraperitoneal injections of ketamine/xylazine for anesthesia. First blood was taken in heparin coated 1 mL syringes by cardiac puncture of the right ventricle and centrifuged at 4000× *g* for 10 min at 4 °C. As done 24 h before to the right kidneys, the left kidneys were cut into two similar sections, whereby one section was fixed in 4% formaldehyde and subsequently embedded in paraffin and the other section was snap frozen in liquid nitrogen for further analyses.

### 4.3. Blood Analyses

Measurements of plasma urea and creatinine were conducted by the central laboratory of the University Hospital of Cologne using a Cobas C 702 (Roche Diagnostics, Mannheim, Penzberg, Germany).

### 4.4. Histopathology

Following the fixation in 4% formaldehyde, all kidney tissues were embedded in paraffin and cut into 2 µm slices. Periodic acid-Schiff (PAS) reaction staining the slices were used for morphological evaluations. One experienced nephropathologist analyzed the tubular damage of each kidney in a blinded fashion by using a composite score as previously described [17,70]. The score reflects the percentage of tubules, which presented epithelial flattening, loss of nucleus, epithelial vacuolization, debris and loss of brush border (0: no change, 1: 1–24%; 2: 25–50%; 3: >50%). The mean sum for each treatment group was plotted.

Immunohistochemical analysis was carried out by staining Anti-cleaved Caspase-3 antibody (Asp175, Cell Signaling Technology, Danvers, MA, USA) in a dilution of 1:400 according to the manufacturer’s protocol. Images were acquired in a 20× magnification using a slidescanner SCN4000 (Leica, Wetzlar, Germany). Beyond that, terminal deoxynucleotidyl transferase dUTP nick end labeling (TUNEL) staining was done using the DeadEnd Fluorometric TUNEL System (Promega, Madison, WI, USA) according to the manufacturer’s protocol.

### 4.5. Image Acquisition

Images were taken by an Axiovert 200M microscope provided with ZEN software (version Zen 2.6, Zeiss, Oberkochen, Germany).

### 4.6. Quantification of Ethanolamides

Levels of ethanolamides in human and murine serum samples were determined by liquid chromatography coupled to electrospray ionization tandem mass spectrometry (LC-ESI-MS/MS) using previously described methods [71,72,73].

Extraction of serum samples was performed as previously described [72] with several modifications: After thawing on ice, 200 µL of serum were diluted with 800 µL of ice-cold PBS. To all samples, 10 µL of an internal standard mixture, containing 100 pg/µL of each of the following deuterated ethanolamides were added: palmitoyl (16:0), ethanolamide-d_4_ oleoyl (18:1), ethanolamide-d_4_ and arachidonoyl (20:4) ethanolamide-d_4_ (Cayman Chemical). After incubation on ice for 15 min, 1 mL of toluene was added. Extraction was performed by shaking the samples in a ThermoMixer (Eppendorf) for 1 min at 4 °C and 2000 rpm. After centrifugation (16,100 RCF, 5 min, 4 °C), the upper (organic) phase was transferred to a new tube and dried under a stream of nitrogen. The dried extract was resolved in 300 µL of methanol/water 3:1 (*v/v*). The samples were again centrifuged, and 100 µL of supernatant were transferred to autoinjector vials.

LC-MS/MS analysis was performed by injection of 35 µL of sample onto a Core-Shell Kinetex C18 column (150 mm × 2.1 mm ID, 2.6 µm particle size, 100 Å pore size, Phenomenex) and detection using a QTRAP 6500 triple quadrupole/linear ion trap mass spectrometer (SCIEX). The LC (1260 Infinity Binary LC System, Agilent Technologies, Santa Clara, CA, USA) was operated at 40 °C and at a flow rate of 0.5 mL/min with a mobile phase of 5 mM ammonium acetate and 0.1% acetic acid in water, pH 3.5 (solvent A) and acetonitrile (solvent B). Ethanolamides were eluted with the following gradient: initial, 10% B; 3 min, 100% B; 7 min, 100% B; 8 min, 90% B; 11 min, 90% B [73].

Ethanolamides were monitored in the positive ion mode with their specific multiple reaction monitoring (MRM) transitions [71]. The instrument settings for nebulizer gas (Gas 1), turbogas (Gas 2), curtain gas, and collision gas were 70 psi, 55 psi, 35 psi, and medium, respectively. The interface heater was on, the Turbo V ESI source temperature was 550 °C, and the ionspray voltage was 5 kV. For all MRM transitions, the values for the entrance potential and the cell exit potential were 10 V and 8 V, respectively. The collision energies ranged from 17 to 25 V, and the declustering potentials from 70 to 110 V.

The LC chromatogram peaks of endogenous ethanolamides and deuterated internal standards were integrated using the Analyst 1.6.3 software (SCIEX, Darmstadt, Germany). Endogenous ethanolamides were quantified on the basis of external calibration curves that were calculated from LC-MS/MS measurements of serially diluted synthetic palmitoyl ethanolamide, oleoyl ethanolamide, and arachidonoyl ethanolamide standard solutions within the range of 0.0 to 40.8 pmol on the column. To each dilution, fixed amounts of the deuterated internal standards were added. The standard calibration curves were plotted based on molar concentration versus peak area ratio of ethanolamide standards to deuterated internal standards. Linearity and correlation coefficients (R2) of the calibration curves were obtained via linear regression analysis. R2 of the calibration curves was >0.99. Linoleoyl (18:2) ethanolamide was quantified on the basis of the calibration curve of oleoyl ethanolamide for the quantification of docosahexaenoyl (22:6) ethanolamide, the calibration curve of arachidonoyl ethanolamide was used. The calculated amounts of endogenous ethanolamides were normalized to the volume of the serum sample.

### 4.7. Human Sample Acquisition

Human samples were obtained as part of three different clinical studies at our center. The first cohort consisted of elderly subjects (>60 years) and was provided with a formula diet with the amount of 60% of daily energy expenditure (DEE) for seven days prior to a coronary-arteria bypass graft ([36], study name: CR_KCH, clinicaltrials.gov identifier: NCT01534364). The second cohort (CR_LSP, clinicaltrials.gov identifier: NCT02745444) consisted of middle-aged (42–64 years, *n* = 6) healthy living kidney donors provided with a formula diet with the amount of 50% of the DEE for the last week prior to kidney donation. The subjects of the third cohort (OptiFast, application number of the local ethics committee: 10–226) were middle-aged (29–65 years, *n* = 8) and obese (BMI > 30 kg/m^2^). They were kept on a formula with the fixed amount of 800 kcal/d for three months. Blood samples included in our analyses were taken at study inclusion and after the diet, after at least 5 h of fasting. All studies were operated in accordance with the Declaration of Helsinki and the good clinical practice guidelines by the International Conference on Harmonization. All patients provided informed consent and approval of each study protocol was obtained from the local institutional review board (Ethics committee of the University of Cologne, Cologne, Germany).

### 4.8. Functional Data and Statistics

For statistical analysis, GraphPad Prism 8 (GraphPad, San Diego, CA, USA) was used. For analyses of mouse studies with three or more treatment groups, one-way analyses of variances (ANOVA) followed by Tukey post-hoc tests were used. For analyses of the longitudinally acquired human blood samples, paired Student’s t-tests were performed. In all cases, two-tailed *p*-values <0.05 were defined as significant and all data are represented as means ± standard deviations (SD).

## Figures and Tables

**Figure 1 ijms-22-05485-f001:**
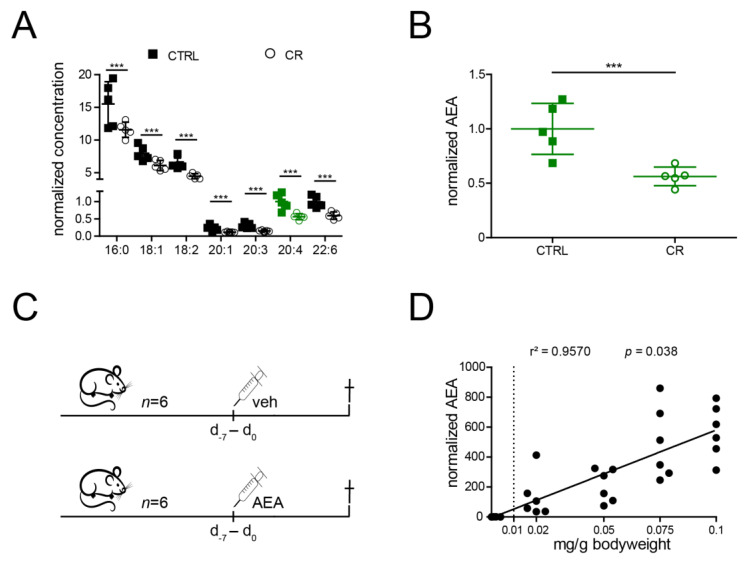
Several NAEs and especially Anandamide are reduced by CR and dose finding reveals a linear Anandamide-increase. (**A**) Levels of the measured set of endocannabinoids on ad libitum diet (CTRL) in comparison to calorically restricted male mice (CR). Concentrations were normalized to the mean AEA levels in ad libitum fed mice (*n* = 5 per group). AEA is labelled in green. (**B**). Enlarged visualization of AEA levels (shown in green in (**A**)) normalized to the mean of ad libitum fed male mice (CTRL) in comparison to calorically restricted male mice (CR) (same values as in A, *n* = 5 per group). (**C**). Schematic illustration of the dose-finding experiment. (**D**) AEA levels after intraperitoneal injections of 0.00 (vehicle), 0.02., 0.05, 0.075, and 0.1 mg/g bodyweight resulted in a linear rise of AEA. AEA concentrations were normalized to the mean of the vehicle-treated mice (*n* = 6 per group). Abb.: 16:0 N-palmitoyl ethanolamide; 18:1: N-oleoyl ethanolamide; 18:2: N-linoleoyl ethanolamide; 20:1: N-eicosenoyl ethanolamide; 20:3: N-eicosatrienoyl ethanolamide; 20:4: N-arachidonoyl ethanolamide/Anandamide (marked in green); 22:6: N-Docosahexaenoyl ethanolamide; AEA: Anandamide; Bars indicate means ± standard deviation; CR: caloric restriction; CTRL: no intervention; d: day; veh: vehicle control; bars indicate mean ± standard deviation; ***: *p*-value <0.001; dashed line: chosen target dose of 0.01 mg/g bodyweight.

**Figure 2 ijms-22-05485-f002:**
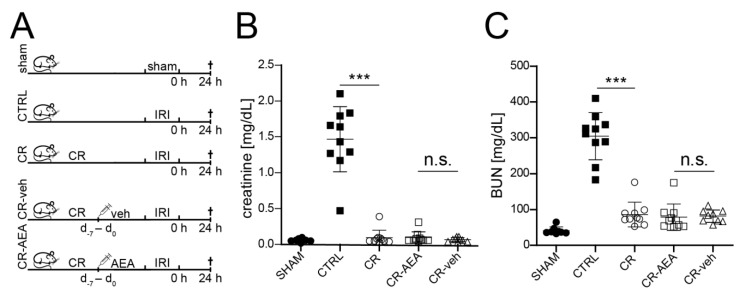
Intraperitoneal AEA administration does not abrogate the CR-mediated kidney protection in mice. (**A**) Schematic illustration of the experimental setup (*n* = 10 per group) (**B**). Plasma creatinine values 24 h after ischemia-reperfusion injury (IRI). (**C**) Plasma blood urea nitrogen values 24 h after IRI. Abb.: CR: caloric restriction; CR-AEA: calorically restricted mice treated with AEA-injections on the last eight days prior to ischemia-reperfusion injury (IRI); CR-veh: calorically restricted mice treated with vehicle-injections on the last eight days prior to ischemia-reperfusion injury (IRI); CTRL: no preconditioning; CTRL-AEA: ad libitum fed mice with AEA-injections on the last eight days prior to ischemia-reperfusion injury (IRI); CTRL-veh: ad libitum fed mice with vehicle-injections on the last eight days prior to ischemia-reperfusion injury; d: day; IRI: ischemia-reperfusion injury; sham: right nephrectomy followed by no-clamping of the left renal pedicle. Bars indicate mean ± standard deviation; ***: *p*-value < 0.001; n.s.: *p*-value > 0.05.

**Figure 3 ijms-22-05485-f003:**
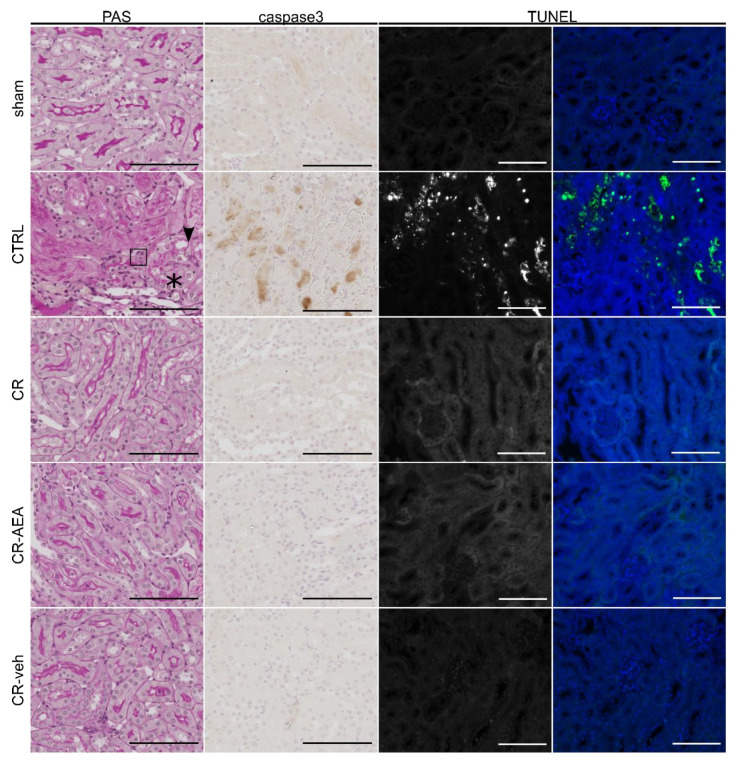
Histological confirmation: intraperitoneal AEA administration does not abrogate the protective effect of CR. Histological analyses reveal higher tubular damage in ad libitum fed animals in comparison to calorically restricted male mice. AEA-injection does not abrogate the protection by CR in mice.Abb.: Asterisk: brush border loss, arrowhead: epithelial flattening; CR: caloric restriction; CR-AEA: calorically restricted mice treated with AEA-injections on the last eight days prior to ischemia-reperfusion injury (IRI); CR-veh: calorically restricted mice treated with vehicle-injections on the last eight days (d) prior to ischemia-reperfusion injury (IRI); CTRL: no preconditioning; d: day; IRI: ischemia-reperfusion injury; sham: right nephrectomy followed by no-clamping of the left renal pedicle; square: pyknosis; PAS: Periodic acid-Schiff reaction; caspase3: immunohistochemical staining against cleaved-caspase3; Scale bar: 100 µm; TUNEL: TdT-mediated dUTP-biotin nick end labeling.

**Figure 4 ijms-22-05485-f004:**
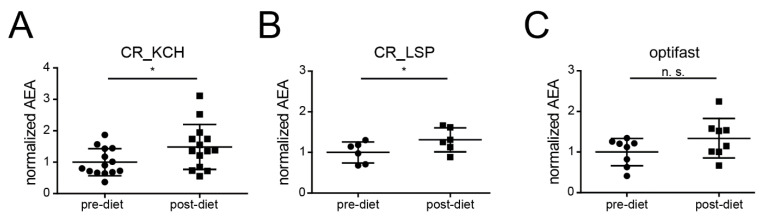
AEA-increase after short-term CR in humans. (**A**) Normalized AEA levels in human serum samples withdrawn from elderly subjects (>60 years) before (pre-diet) and after dietary intervention (post-diet) with a formula diet consisting of the amount of 60% of daily energy expenditure (DEE) for seven days prior to a coronary-arteria bypass graft. (**B**) Normalized AEA levels in human serum samples withdrawn from middle-aged (42–64 years, *n* = 8) healthy living kidney donors before (pre-diet) and after dietary intervention (post-diet) with formula diet with the amount of 50% of the DEE for the last week prior to kidney donation. (**C**) Normalized AEA levels in human serum samples withdrawn from middle-aged (29–65 years, *n* = 8) obese (BMI > 30 kg/m^2^) subjects before (pre-diet) and after dietary intervention (post-diet) with the fixed amount of 800 kcal/d for 3 months. Abb.: AEA (Anandamide) concentrations were normalized to the mean of the pre-diet concentrations. Depicted are means ± standard deviation; *: *p*-value < 0.05, n.s.: not significant (*p*-value > 0.05).

## Data Availability

All data in this study can be obtained from the authors.

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
