# Peer review of "Modulation of Endocannabinoids by Caloric Restriction Is Conserved in Mice but Is Not Required for Protection from Acute Kidney Injury"

_ijms, 2021, doi:10.3390/ijms22115485_

Round 1
Reviewer 1 Report
In this interesting article authors show that contrary to hypothesis derived from C. Elegans , caloric restriction in mice seems to protect from experimental AKI induced by ischemia-reperfusion independently from endocannabinoids. Likewise, in humans caloric restriction does not seem to reduce endocannabinoids but rather increase them.
My main comments relate to the relatively poor presentation and valorization of the authors works to clearly improve.
Author Response
We thank the editors and reviewers for their helpful evaluation of our manuscript.
Reviewer(s)' Comments to the Authors:
Reviewer: 1
Comments and Suggestions for Authors
In this interesting article authors show that contrary to hypothesis derived from C. Elegans, caloric restriction in mice seems to protect from experimental AKI induced by ischemia-reperfusion independently from endocannabinoids. Likewise, in humans caloric restriction does not seem to reduce endocannabinoids but rather increase them.
My main comments relate to the relatively poor presentation and valorization of the authors works to clearly improve.
Response: We thank the reviewer for this assessment and have adapted the presentation of the data. We have now added more detailed information concerning the general definition of acute kidney injury, preconditioning strategies in humans and a detailed description of NAEs to the introduction. Furthermore we rearranged the presentation of our results:
In detail: we integrated Fig. S1 into Fig. 1 and replaced the bar plots by scatterplots. Furthermore, we rearranged the scale of the x-axis in Fig. 1 D. For a more detailed depiction of the histological results, we rearranged and enlarged the panels of Fig.3 and Fig.S4 and increased the magnification of the image sections. Additionally, we added the size of treatment groups to the figure legends. Finally, we added more references for the methods used for the quantification of ethanolamides.

Reviewer 2 Report
The authors investigate : Modulation of endocannabinoids by caloric restriction in mice and its links to acute kidney injury.
Comments:
Paragraph 1 -In the introduction a general definition of AKI should be provided
Paragraph 2- Why have therapeutics in rats been successful but not humans?
In the introduction more description of N-acylethanolamines is needed. There seems to be no justification in the introduction why this molecule was investigated? How is it made? What is a potential mechanism of action?
IN general, I believe if a p value is >0.05 you just report it as not signficiant.
I believe S1 should be in the paper-not a supplement
This may just be an issue with the format for the manuscript submission but can Figure 3 be enlarged? Details are difficult to see.
Figure 4-for optifast it is not significant so no P value is needed
I am not clear on the relevance of the obese patients? Obesity is not a risk factor for AKI so why was it used?
Lines 196-200 should be in the introduction, same for 212-222
Paragraph starting at 224 suggests that this should have been investigated in your model? Why was it not?
Methods-I think statistics should be the last section of the methods
Have the authors or others published the method for Quantification of ethanolamides-can these be cited to ensure they are valid?
Author Response
We thank the editors and reviewers for their helpful evaluation of our manuscript.
Reviewer: 2
Comments and Suggestions for Authors
The authors investigate : Modulation of endocannabinoids by caloric restriction in mice and its links to acute kidney injury.
Comments:
Paragraph 1 -In the introduction a general definition of AKI should be provided.
Response: We have added a general definition of AKI in the beginning of the introduction.
Paragraph 2- Why have therapeutics in rats been successful but not humans?
Response: Thank you for giving us the opportunity to highlight some potential barriers hampering transferability of caloric restriction (CR) to the clinic. From our point of view, there are several explanations why preconditioning strategies in humans are not as successful as in rodents, some of which we recognized in our own recently published clinical trials. Patients at risk are usually quite heterogenous cohorts due to numerous co-morbidities and age. Lack of knowledge regarding normal food and calorie intake especially in elderly multimorbid patients makes trial design complicated. Concealed adherence problems may occur in the CR group due to difficulties regarding food intake surveillance on the one hand. On the other hand, unintended dietary restriction may occur in the control group due to expectations regarding outcome improvement. Additionally, blinding is often difficult in trials examining dietary interventions. Despite the fact that this is enabled when using formula diets, a reduction in caloric intake may easily noticed by the participants as a consequence the sensation of hunger. Moreover, recruitment of patients may be limited by lack of motivation to restrict dietary habits before a surgical intervention. Besides, safety concerns, especially in elderly, frail and severely-ill patients, may play a role with prolonged CR.
Consequently, increasing our knowledge on the molecular mechanisms underlying the power of CR in rodents will be crucial to clinical translation.
We now explore these points in the introduction.
In the introduction more description of N-acylethanolamines is needed. There seems to be no justification in the introduction why this molecule was investigated? How is it made? What is a potential mechanism of action?
Response: Thank you for this advice, we now set forth this paragraph in greater detail. NAEs are endocannabinoids and as such lipid-derived signaling molecules that are – among other organs - synthesized in the kidney. NAEs are formed within the tissues by N-acyl-ethanolamine phospholipids (NAPE) and can activate the G-protein-coupled receptors CB-1 and CB-2 in e.g. the kidney. They are involved in the regulation of food intake, energy metabolism, inflammation and vascular tonus. A decrease of NAEs was required and sufficient for the CR-mediated extension of lifespan in nematodes.
IN general, I believe if a p value is >0.05 you just report it as not signficiant.
Response: Thank you. We replaced this.
I believe S1 should be in the paper-not a supplement
Response: Thank you for this advice. We integrated Fig S1 to Fig. 1A.
This may just be an issue with the format for the manuscript submission but can Figure 3 be enlarged? Details are difficult to see.
Response: This is an important remark, thank you. We rearranged and enlarged the panels in Fig.3 and FigS4and increased the magnification of the image section.
Figure 4-for optifast it is not significant so no P value is needed
Response: Thank you. We replaced it by not significant (n.s.).
I am not clear on the relevance of the obese patients? Obesity is not a risk factor for AKI so why was it used?
Response: Indeed, obesity was the subject of investigation. This is just a description of the cohort. This cohort underwent CR which is the reason why we used the resulting biosamples for our analyses.
Lines 196-200 should be in the introduction, same for 212-222
Response: We inserted parts of the mentioned lines in the introduction.
Paragraph starting at 224 suggests that this should have been investigated in your model? Why was it not? (Paragraph starting at 224: Current mechanistic concepts suggest, that CR-mediated cellular resistance is based on reduced IGF-1/Insulin signaling [18], activation of AMPK [47] and furthermore an inhibition of the mammalian target of rapamycin (mTOR) [48]. In the nematode, Lucanic et al. concluded,that the TOR-pathway might control NAE levels in response to nutrient intake and that reducedNAE signaling is a component of CR-mediated lifespan extension. On the other hand,there is evidence that NAEs activate mTOR [49, 50] and AMPK signaling in renal proximal tubular cells [51] as well as hepatocytes [52].)
Response: Our current analysis revealed that contrary to the hypothesis based on data in the nematode model, caloric restriction in mice seems to protect from experimental AKI induced by ischemia-reperfusion independently from endocannabinoids. Furthermore, our data show that reduction of endocannabinoids by CR appears to occur in rodents but not to be conserved in humans. Consequently, we did not examine mechanisms involved in NAE signaling. We have discussed possible connections in the discussion, yet these mechanisms need further evaluation even in other models and settings.
Methods-I think statistics should be the last section of the methods
Response: Fixed.
Have the authors or others published the method for Quantification of ethanolamides-can these be cited to ensure they are valid?
Response: Thank you for this comment. Of course, we used a valid method that has been published before. We added the references in the first sentence of the paragraph of the methods.

Round 2
Reviewer 1 Report
This revised version has been clearly and deeply improved for the readers